

# Improving decolorization of dyes by laccase from *Bacillus licheniformis* by random and site-directed mutagenesis

Tongliang Bu[*], Rui Yang[*], YanJun Zhang, Yuntao Cai, Zizhong Tang, Chenglei Li, Qi Wu and Hui Chen

College of Life Science, Sichuan Agricultural University, Ya'an, China
[*] These authors contributed equally to this work.

## ABSTRACT

**Background**. Dye wastewater increases cancer risk in humans. For the treatment of dyestuffs, biodegradation has the advantages of economy, high efficiency, and environmental protection compared with traditional physical and chemical methods. Laccase is the best candidate for dye degradation because of its multiple substrates and pollution-free products.

**Methods**. Here, we modified the laccase gene of *Bacillus licheniformis* by error-prone PCR and site-directed mutagenesis and expressed in *E. coli*. The protein was purified by His-tagged protein purification kit. We tested the enzymatic properties of wild type and mutant laccase by single factor test, and further evaluated the decolorization ability of laccase to acid violet, alphazurine A, and methyl orange by spectrophotometry.

**Results**. Mutant laccase Lac[ep69] and D500G were superior to wild type laccase in enzyme activity, stability, and decolorization ability. Moreover, the laccase D500G obtained by site-directed mutagenesis had higher enzyme activity in both, and the specific activity of the purified enzyme was as high as 426.13 U/mg. Also, D500G has a higher optimum temperature of 70 °C and temperature stability, while it has a more neutral pH 4.5 and pH stability. D500G had the maximum enzyme activity at a copper ion concentration of 12 mM. The results of decolorization experiments showed that D500G had a strong overall decolorization ability, with a lower decolorization rate of 18% for methyl orange and a higher decolorization rate of 78% for acid violet.

**Conclusion**. Compared with the wild type laccase, the enzyme activity of D500G was significantly increased. At the same time, it has obvious advantages in the decolorization effect of different dyes. Also, the advantages of temperature and pH stability increase its tolerance to the environment of dye wastewater.

Corresponding author
Tongliang Bu, 13941@sicau.edu.cn

# INTRODUCTION

Laccases (phenol-oxygen oxidoreductase; EC 1.10.3.2), a copper-containing polyphenol oxidase, belong to the superfamily of blue poly copper oxidases (MCOs) (*Morozova et al., 2007*; *Hakulinen & Rouvinen, 2015*). Laccase was first discovered in the permeate of *Rhus vernicifera* (*Legrand & Martin, 1958*). Thereafter, they were also found in plants,

fungi, bacteria, and insects (*Chakroun et al., 2010*; *Forootanfar et al., 2011*; *Halaburgi et al., 2011*). There are many studies on fungal laccase and bacterial laccase. White rot fungi of the basidiomycete family are mostly studied in laccase-producing fungi. At present, it has been found that bacterial laccase mainly comes from *Bacillus* sp. (*Mollania et al., 2011*; *Chen et al., 2017*), Streptomyces (*Freeman et al., 1993*), and Pseudomonas (*Francis & Tebo, 2001*). Laccases can catalyze phenols, polyphenols (*Koschorreck et al., 2008*; *Zeng et al., 2011*; *Revanth, Niranjan & Sarma, 2020*), polycyclic aromatic hydrocarbons, certain inorganic substances, and more. As a result, they are widely used for the decolorization of synthetic dyes (*Pereira et al., 2009*; *Mendes et al., 2011*), synthesis of organic substances, food processing, biosensor (*Zhang et al., 2019*), and other fields. The molecular structure of laccase contains four copper ions. There are three types based on magnetic and spectral properties: Type 1 (T1), Type 2 (T2), and type 3 (T3) copper ions. T1 $Cu^{2+}$, located at the substrate-binding site, is responsible for transferring substrate electrons (*Martins et al., 2015*). T2 $Cu^{2+}$ and T3 $Cu^{2+}$ are located at the oxygen molecule binding site, where oxygen molecules combine with electrons to generate water (*Sakurai & Kataoka, 2007*; *Sakurai & Kataoka, 2010*).

Dye wastewater has become one of the main hazardous industrial sewage due to a large number of dyes and intermediates. According to chemical properties, dyes are divided into reactive dyes, acid dyes, basic dyes, disperse dyes, vat dyes, sulfur dyes, mordants, direct dyes, naphthol dyes, solvent dyes, and organic pigments (*Bhatia et al., 2017*). Synthetic dyes with strong carcinogenic polycyclic aromatic hydrocarbons as raw materials have become more commonly used dyes in the printing and dyeing industry because of their stable physical and chemical properties and low cost. Wastewater from the printing and dyeing industry is discharged into freshwater without treatment, which seriously affects the growth of aquatic organisms and microorganisms (*Mishra & Maiti, 2018*), and destroys the self-purification of water bodies (*Tkaczyk, Mitrowska & Posyniak, 2020*; *Gowri, Vijayarghavan & Meenambigai, 2014*). At the same time, azo and anthraquinone dyes will produce a variety of carcinogenic aromatic amines during specific decomposition, which can cause cancer, mutagenesis, and reproductive toxicity (*Ali et al., 2019*). The objects of this study are both azo (methyl orange, alphazurine A) and anthraquinone (acid violet) dyes that are acid.

In terms of dye degradation, white rot fungal laccase has many problems, such as a long culture period and high cost. Therefore, researchers turned their attention to bacterial laccase. (*Michniewicz et al., 2008*; *Hadibarata et al., 2012*; *Tian et al., 2014*; *Zheng et al., 2017*; *Legerská, Chmelová & Ondrejovič, 2018*). In 1993, Givaudan first detected laccase activity in *Awspirillum lipoferum* (*Givaudan et al., 1993*). Numerous studies have shown that certain bacterial laccases, such as CotA laccases from Bacillus capsid protein, are more tolerant than fungal laccase in neutral or alkaline environments. (*Zhang et al., 2012*; *Lu et al., 2012*; *Guan et al., 2014*; *Martins et al., 2015*; *Wang et al., 2016*). This bacterial laccase is more suitable for the environment of dye wastewater and exhibits higher enzyme activity. However, the enzyme activity of a crude enzyme solution of wild bacterial laccase is often lower. Direct modification of proteins is an effective way to improve enzyme activity and stability (*Chen et al., 2017*). This technology mainly includes two strategies: rational design

**Table 1  Strains and plasmids.**

| Strains/plasmids | Genotype/description | Source |
|---|---|---|
| pET-*Lac* | A recombinant plasmid containing with wild gene *Lac* | Laboratory |
| pET-*Lac*[ep69] | A recombinant plasmid containing with mutant gene *Lac*[ep69] | Laboratory |
| pET-*D500G* | A recombinant plasmid containing with mutant gene *D500G* | Laboratory |
| *E. coli* DH5α | Competent cells of plasmid cloning host bacteria | Takara |
| *E. coli* BL21 | Competent cells of plasmid expression host bacteria | Takara |
| pMD19-T | Cloning vector | Takara |
| pET-30b (+) | Expression vector | Takara |
| FDM | Competent cells of plasmid cloning host bacteria | Tiangen |

and directed evolution. Rational designs are usually based on computer-aided structural modeling of enzyme proteins, using site-directed mutation techniques, knockout, and insertions of protein sequences to alter the properties and functions of the target protein. Finally, target protein properties were analyzed by measuring enzyme activity. Directed evolution aims to construct a set of random gene transformations in vitro by mimicking natural evolution, and then select target proteins through library construction and high-throughput screening. It can be achieved by error-prone PCR and DNA recombination techniques (*Bornscheuer & Pohl, 2001*; *Berman, 2008*).

Here, we used a laboratory-constructed plasmid containing laccase constructed by *Bacillus licheniformis* as a template, modified target genes by error-prone PCR and site-directed mutation, screened mutants with higher enzyme activity, and further evaluated the decolorization ability of wild-type and mutant laccase to dyes.

# MATERIALS AND METHODS

## Materials
### Strains and Plasmid
The sources of bacterial strains, vectors, and engineered bacteria used in this article were shown in Table 1. All strains besides cold-induced expression were grown in Luria-Bertani (LB) medium at 37 °C.

### Chemicals and apparatus
Diammonium 2,2′-azino-bis (3-ethylbenzothiazoline-6-sulfonic sulfonate, ABTS), acid violet, alphazurine A, and methyl orange were purchased from Sigma-Aldrich (St. Louis, MO, USA). Restriction enzymes *EcoR* I, *Kpn* I, Prime STAR Max, DNA Marker, and Protein Marker all purchased from TaKaRa (Dalian China). The bacterial genomic DNA extraction kit, instant error-prone PCR kit, agarose gel DNA recovery kit, plasmid extraction kit, and fast site-directed mutagenesis kit all purchased from TianGen (Beijing, China). Ampicillin, kanamycin, and IPTG were purchased from Solarbio (Beijing, China). His-tagged protein purification kit was purchased from Kangweishiji (Beijing, China). All other chemicals were standard reagent grade.

**Table 2  Primers for PCR.**

| Primer | Sequence (5′→3′) |
|---|---|
| Lac-M-F | GACAGCCCAGATCTGGGGTACCATGAAACTTGAAAAATTCGTTGACC |
| Lac-M-R | TTGTCGACGGAGCTCGAATTCTTATTGATGACGAACATCTGTCACTT |
| Lac-M-1-R | CAAAGATTCTCGTATTGGAGGCGT |
| Lac-M-2-F | ATACTGAACGCCTCCAATACGA |
| Lac-D-F | CCTTGTCGACGAAGATTACGGTATGATGCGC |
| Lac-D-R | CCGTAATCTTCCAACTCAAGGATGTGGCAGT |

PCR instrument (T100$^{TM}$ Thermal Cycler, Bio-Rad) Molecular Devices Spectra Max M2 microplate reader, Ultrasonic processor type ultrasonic crusher, E-201-C type pH composite electrode, ZWY-211C constant temperature, Constant temperature incubator, Nucleic acid electrophoresis equipment, Protein electrophoresis equipment, and other equipment. digital pH meter (PHS-25, Shanghai Instrument and Electric Scientific Instrument Co., Ltd.), E-201-C type pH composite electrode.

## Enzyme activity assay

Laccase oxidation reaction was carried out at 50 °C in vitro, using ABTS as a substrate. In a solution containing 50 mM tartaric acid buffer (pH 3.0), 1 mM ABTS and 1 mM CuCl$_2$ 300 µL of enzyme solution was added to three mL of reaction system. After 5 min of reaction, the absorbance at 420 nm was measured. Under the above conditions, the amount of enzyme required to oxidize 1 µmol of ABTS per minute is defined as one unit of enzyme activity (U). The assay was performed in quadruplicate. According to the study of Lu et al., some modifications were made to the enzyme activity determination method (*Lu et al., 2012*).

The formula for calculating enzyme activity is shown in Eq. (1).

$$U(U/mL) = \frac{\Delta A}{\varepsilon bt} \times \frac{V_1}{V_2} \times n \times 10^6 \tag{1}$$

where $\Delta A$ represents absorbance change value in time (t), b is the thickness of cuvette (cm), and t is the reaction time (min). $V_1$ and $V_2$ represent the volume of the reaction system (L) and a crude enzyme solution (L), respectively. n is a dilution ratio. ABTS: $\varepsilon_{420} = 36000$ $\mu^{-1}$ cm$^{-1}$.

## Error-Prone PCR

The laccase gene of *B. licheniformis* was randomly mutated by using the constructed recombinant plasmid pET-*Lac* containing the *Lac* gene as the template and adding the specific primers containing the enzyme cutting site (Table 2). (Note: The error-prone PCR kit is only suitable for reactions with a gene length of less than 1,000 bp, but the Lac gene is 1,542 bp, so segmented PCR is performed). The mutant laccase gene was ligated with the linearized vector pET-30b (+) to form the recombinant plasmid pET-*Lac$^{ep}$*, then transformed into *E. coli* BL21 competent cells, and then the positive transformants were
screened with LB medium containing ampicillin. Using ABTS as a substrate, a 96-well plate method was used to screen mutants with laccase activity. Positive clones were sequentially added to 96-well plates for overnight culture, in which 200 µL of resistance medium was added to the wells beforehand. The 10 µL culture was added to a new 96-well plate in turn and incubated for 4 h using IPTG induction. After the culture, the bacteria were precipitated at −80 °C and repeatedly frozen and thawed three times. The bacteria were resuspended with Tri-HCl and lysed with lysozyme, and the lysate was centrifuged to obtain the supernatant. Enzyme activity of the supernatant was determined and mutants with higher enzyme activity were screened.

## Site-directed mutagenesis

Forward and reverse mutation primers Lac-D-F and Lac-D-R were designed using the fast site-directed mutagenesis kit. Using plasmid pET-*Lac* containing *Lac* as a template amplified mutant Lac. PCR products were digested using 1 µL *Dpn* I enzyme (20 U/ µL) at 37 °C for 1 h. The *Dpn* I enzyme can digest methylated templates. After treatment with *Dpn* I enzyme, it was then transformed into FDM competent cells. All mutants were screened and further confirmed by DNA sequencing. Plasmids containing the desired mutations were then transformed into *E. coli* BL21 (DE3) for protein expression.

## Expression and Purification of laccases

*E. coli* BL21 (DE3) containing WT laccases gene (*Lac*) and mutant laccase genes (*Lac*[ep69] and *D500G*) were cultured overnight in 10 mL LB medium containing ampicillin (100 mg/mL) at 37 °C with shaking (180 rpm). Afterward, the overnight pre-culture was inoculated into a fresh 50 mL culture medium (1% inoculation) containing ampicillin (100 mg/mL) and incubated at 37 °C with shaking (180 rpm) until an optical density at 600 nm ($OD_{600}$) of 0.6 was reached. Then, isopropyl-β-D-1-thiogalactopyranoside (IPTG) was added to the culture medium to a final concentration of 0.1 mM, and the culture was induced at 16 °C, 100 rpm. Samples were taken every two hours and 12% SDS-PAGE was used to detect the expression of target proteins (*Laemmli, 1970*). Meanwhile, *E. coli* BL21 cells containing empty vector pET-30b (+) were used as control. Centrifuge (10 min, 8, 000× g, 4 °C) to collect induced bacterial cells. Cells were crushed on ice and centrifuged (20 min, 8, 000× g, 4 °C) to remove cell debris. Then, the supernatant was treated at 70 °C for 15 min, and the denatured protein was removed by centrifugation (10 min, 10, 000× g, 4 °C) (*Koschorreck, Schmid & Urlacher, 2009*; *Nasoohi et al., 2013*).

Based on the histidine tag carried on the vector pET-30b (+), we purified the recombinant laccase using the His-tagged protein purification kit. Then, according to the SDS-PAGE results, the pure enzyme solution was desalted by Amicon ultrafiltration (membrane retention value: 10 kDa; Millipore, Billerica, Ma, USA). According to Bradford's method, bovine serum albumin was used to make the protein standard curve, and the protein content of purified laccase was determined (*Bradford, 1976*).

## Characterization of laccases

### Optimum temperature and thermal stability

Optimum temperature and temperature stability of purified laccase activity were assessed by relative enzyme activity measured at different temperatures. The enzyme activity of laccase at different temperatures was measured after incubation at 50, 55, 60, 65, 70, 75, 80, 85, 90, 95, and 100 °C for 1 h under standard conditions, and the enzyme activity measured at 4 °C was defined as 100%.

### Optimum pH and pH stability

Optimum pH and pH stability of purified laccase activity were assessed by relative enzyme activity measured at pH 2.0, 2.5, 3.0, 3.5, 4.0, 4.5, 5.0, 5.5, 6.0, 6.5 under standard conditions, respectively. The pH of the enzyme reaction system was adjusted by 0.05 M citric acid-$Na_2HPO_4$ buffer to 2.0, 2.5, 3.0, 3.5, 4.0, 4.5, 5.0, 5.5, 6.0, 6.5, and incubated at 4 °C overnight to measure the enzyme activity at different pH. The enzyme activity measured under pH 4 was defined as 100%, and the relative enzyme activity was calculated.

### Copper ion concentration

To detect the effect of copper ion concentration on laccase activity, $CuCl_2$ solution with a final concentration of 0, 0.2, 0.4, 0.6, 0.8, 1.0, 1.2, 1.4, 1.6, 1.8, 2.0 mM was added to the reaction system, and laccase activity was determined under the optimum reaction conditions. The relative enzyme activity was calculated by taking the laccase enzyme activity measured under the same reaction system without adding $CuCl_2$ solution was determined to be 100%. The data were processed and analyzed using Origin 8.0 software. The decolorization rate calculation formula is shown in Eq. (2).

$$D = \frac{A_0 - A_1}{A_0} \times 100\% \tag{2}$$

where D represents the decolorization rate (%), $A_0$ is the initial absorbance of the dye solution at the maximum absorption wavelength, $A_1$ results from the initial absorbance of the dye solution at the maximum absorption wavelength after the reaction.

## Dye decolorization

The decolorization ability of Lac, $Lac^{ep69}$, and D500G was tested with three dyes of acid violet ($\lambda_{max}$=600 nm), alphazurine A ($\lambda_{max}$=637 nm), and methyl orange ($\lambda_{max}$=470 nm). Add 100 mg/mL purified enzyme protein, dye (40 mg/L acid violet, 20 mg/L alphazurine A, 20 mg/L methyl orange) and 1 mM $CuCl_2$ to tartaric acid buffer (0.1 M, pH 4.0) for decolorization reaction. Samples were taken and centrifuged regularly (12,000 rpm, 2 min), and the decolorization effect was determined by spectrophotometry. All reactions are performed in quadruplicate.

## Bioinformatics analysis

Using DANAMAN, ProtParam (http://web.expasy.org/protparam/), SOPMA (http://www.expasy.ch/swissmod/SWISS-MODEL.heml), and Swiss-Model (http://www.expasy.ch/swissmod/SWISS-MODEL.heml) tools for gene sequence comparison, basic property analysis, secondary structure prediction, and tertiary structure prediction of laccases from wild-type and mutant strains, respectively.
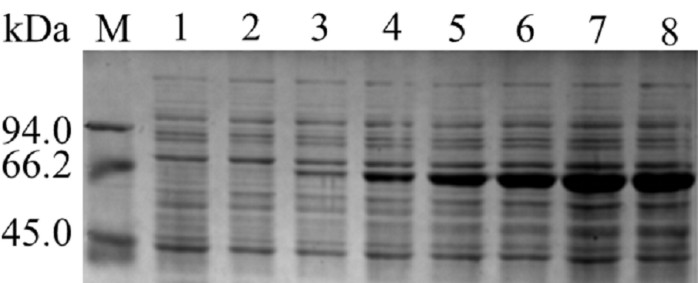

**Figure 1** **SDS-PAGE analysis of pET-** *Lac.* Images of strips stained with Coomassie Brilliant Blue R250. M: Marker; Lane 1: negative control; Lane 2–8: The products of induction 0 h, 2 h, 4 h, 6 h, 8 h, 10 h, 12 h.

## Statistical analysis

Quadruplicate in experiments was conducted on each sample to ensure good repeatability. Statistical Program of Social Science (SPSS 17.0, Chicago, IL, USA) software was used for one-way analysis of variance (ANOVA) by using Duncan's test of the data to complete the significant difference test, at a significant level of $P < 0.05$.

## RESULTS

### Expression and purification of wild-type and mutant laccases

Nucleotide sequencing of the plasmid extracted from the positive transformant confirmed that *Lac*'s ORF contains 1,542 bp that theoretically encode 513 amino acids with a molecular weight of about 60 kDa. (Data S1) The protein sequence has a 99% identity to the *Lac* gene of *B. licheniformis* (MK427697.1).

The results of SDS-PAGE showed that the enzyme expression reached the highest level, after 10 h of IPTG induction. Because the expression level increases from 2 to10 h, 10 to 12 h tends to remain unchanged. The results were shown in Fig. 1. After 10 h of induction, the cells were sonicated and purified. SDS-PAGE electrophoresis results of the purified product, intracellular supernatant, and cell debris are shown in Fig. 2. We found that under low temperature and low concentration inducer conditions, the recombinant Lac was mainly expressed in a soluble state but has a low content in the precipitate. And after purification, a single band can be obtained (Lane 1). Using ABTS as the substrate, the specific activity of the purified Lac was 121.75 U/mg.

Here, we used a 96-well plate method to screen a mutant strain Lac[ep69] with higher enzyme activity than Lac, from the mutated Lac gene library generated by error-prone PCR. The mutant strain Lac[ep69]'s specific activity is 51.24 U/mg, which was 1.35 times higher than that of the wild strain 37.84 U/mg. After the alignment of multiple laccase gene sequences, we found a conserved aspartic acid at position 500 of the laccase, replacing aspartic acid with glycine to carry out the targeted modification of laccase (The data isn't displayed). The specific activity of Lac[ep69] was 170.45 U/mg, and D500G was 426.13 U/mg.
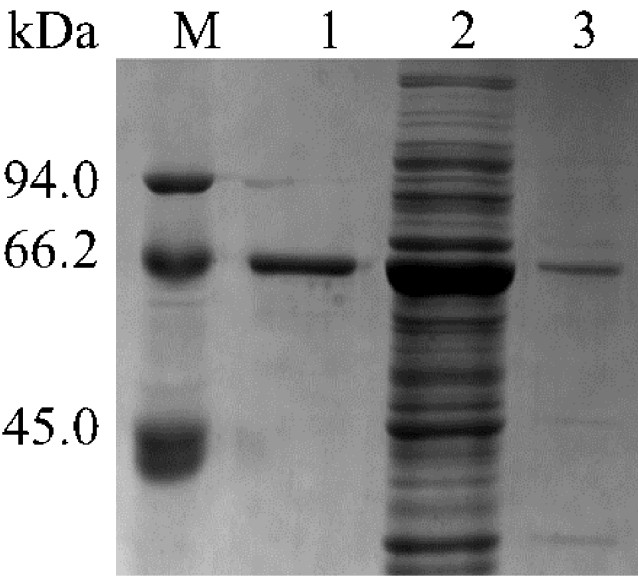

**Figure 2  Detection of Lac purified SDS-PAGE.** M: Marker; Lane 1: Purified product; Lane 2: intracellular supernatant; Lane 3: cell debris.

## Characterization of the purified wild-type and mutated laccases
### Optimal pH and pH stability

The optimized pH and pH stability results of Lac, Lac[ep69], D500G after purification are shown in Figs. 3A, 3B. Lac[ep69] and Lac have the same optimal pH of 4.0, while D500G is more neutral than Lac, with a pH increase of 0.5. Figure 3B shows that at a pH of 4.0–5.5, the relative enzyme activity of Lac[ep69] after 1 h of incubation is stable above 75%, while the relative enzyme activity of D500G after 1 h of incubation at 4.5–6.5 is stable above 80%. In contrast, D500G has higher enzyme activity and stability in a neutral environment.

### Optimum temperature and temperature stability

Tested the optimal temperature and temperature stability of Lac, Lac[ep69], D500G results as shown in Figs. 3C, 3D. The optimum temperature for both Lac[ep69] and D500G was higher than Lac, which was 80 and 70 °C respectively. Their temperature stability is high. After incubating at 50−80 °C for 1 h, Lac[ep69] enzyme activity remained above 80%, while D500G enzyme activity remained above 85%. It indicated that mutants were more tolerant of temperature than the wild type, had higher enzyme activity in higher temperature environments, and were more suitable for enzyme industrialization. Besides, when the temperature increased to 85 °C the enzyme activity of Lac, Lac[ep69], and D500G showed a rapid decline trend, indicating that excessive temperature denatured the enzyme and inactivated it.

### Copper ion concentration

Tested the effect of different concentrations of copper ions on enzyme activity. As shown in Fig. 4, when the concentration of copper ions in the reaction system was 12 mM, the activity of wild-type laccase reached its peak. However, the mutant strains Lac[ep69] and

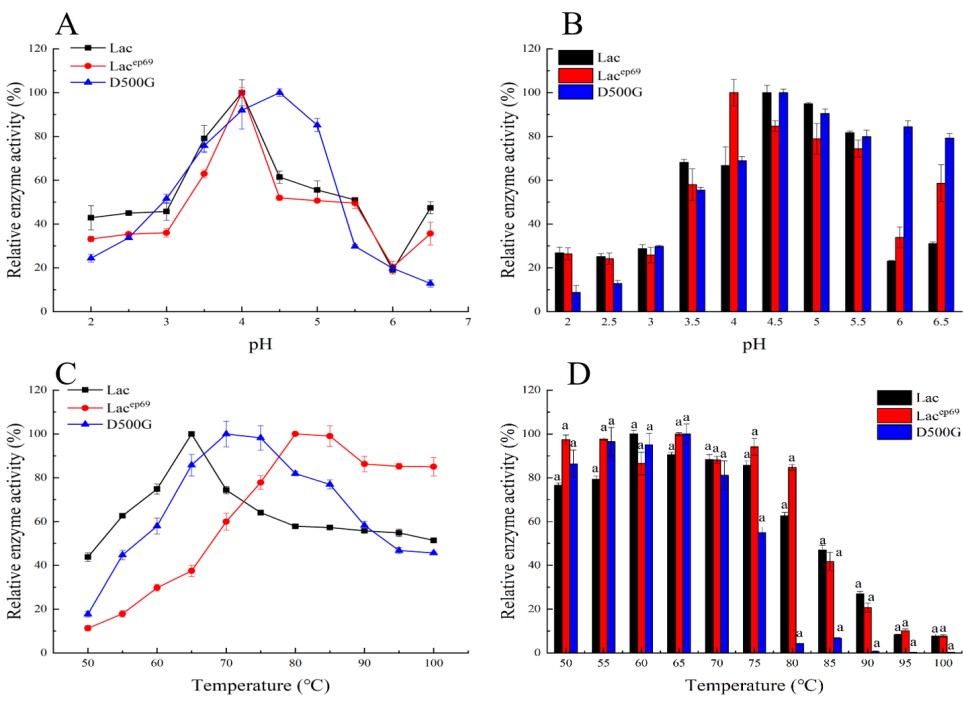

**Figure 3 Optimum pH, temperature, and stability of engineering bacteria and mutants.** (A) Optimum pH; (B) pH stability; (C) optimum temperature; (D) temperature stability.

D500G only need 10 mM copper ion concentration to achieve maximum enzyme activity. This indicated that the mutant was more sensitive to copper ions than the wild type Lac.

### Km value determination

$K$m represents the magnitude of the affinity of the enzyme to the substrate. The larger the $K$m, the smaller the affinity of the enzyme and the substrate, the lower the enzyme activity. We used Origin8.0 software to predict the km of Lac, Lac[ep69], and D500G after purification. The $K$m value of engineered bacteria Lac, mutant strains Lac[ep69] and D500G was 21.69 mM, 19.54 mM, and 10.50 mM respectively, the trends were consistent with the previous experimental of the enzyme protein activity determination, which showed that the increase in affinity leads to the increase in enzyme activity.

### Dye decolorization assays by laccase

Figures 5A–5C show the decolorization of dyes by within 6 h. First, the rapid reaction phases of Lac, Lac[ep69], and D500G were all 0–1 h. Secondly, the degradation rates of the two mutant enzymes were significantly better than those of the wild type. And D500G has the best decolorization ability. The decolorization rate of Lac to three dyes: methyl orange 15%, alphazurine A 30%, and acid violet 40%. The decolorization rate of D500G for three dyes: methyl orange 18%, alphazurine A 70%, and acid violet 78%.

As shown in Figs. 5A–5C, the enzymatic reaction within 0–1 h belongs to the rapid reaction period, and the enzyme activity is relatively less affected at this time. Therefore, we compared the decolorization of the three dyes by the wild type and the mutant within
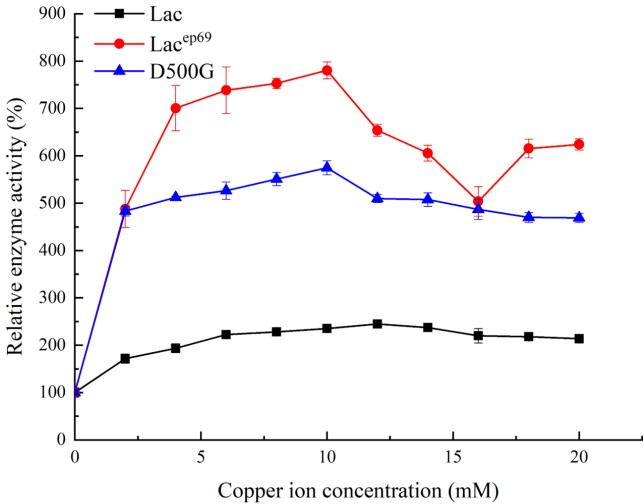

**Figure 4 Effect of copper ion concentration on wild type and mutant laccase.**

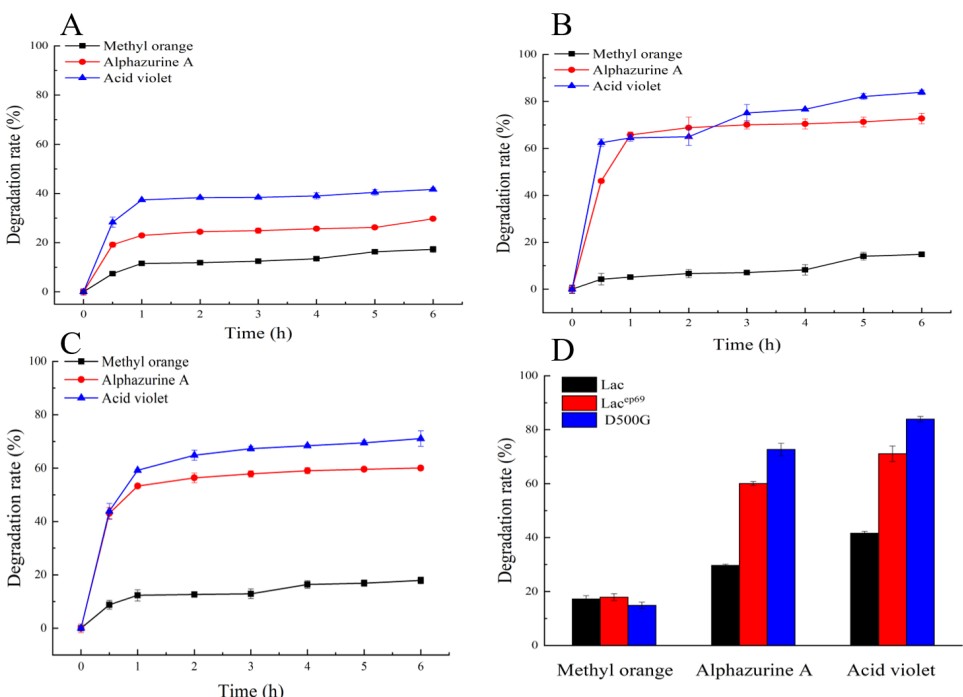

**Figure 5 Degradation of dyes by different laccases.** (A, B, C) Decolorization of dyes by purified Lac, Lac[ep69], D500G at different time. (D) Decolorization of dyes by purified Lac and its mutants Lac[ep69] and D500G in 1 h.

**Table 3  Physicochemical properties of Lac, Lac$^{ep69}$ and D500G.**

| Strain | Molecular weight (Da) | Isoelectric point | Positively charged residues | Negatively charged residues | Instability coefficient |
|---|---|---|---|---|---|
| Lac | 59074.20 | 6.25 | 60 (Arg+Lys) | 70 (Asp+Glu) | 40.03 |
| Lac$^{ep69}$ | 59061.11 | 6.25 | 60 (Arg+Lys) | 70 (Asp+Glu) | 39.57 |
| D500G | 59016.17 | 6.31 | 60 (Arg+Lys) | 70 (Asp+Glu) | 39.40 |

1 h. The result is shown in Fig. 5D. There are differences in the decolorization rates of different dyes by laccase. Lac, Lac$^{ep69}$, and D500G all have lower degradation rates for methyl orange, but higher degradation rates for alphazurine A and acid violet. For the three dyes, the degradation efficiency of the mutant is always better than that of the wild type. Especially D500G shows a significantly enhanced degradation rate for all three dyes. From the perspective of enzyme activity, the higher enzyme activity of D500G provides the possibility for dye degradation.

## Analysis of laccase protein structure

The results of the primary structure show that compared with the original Lac, the mutant strain Lac$^{ep69}$ has six pairs of base changes (T811C, A944G, T998C, T1303C, A1389C, and T1440A) in the sequence. Correspondingly, the six pairs of amino acids (Cys271Arg, Lys315Arg, Ile333Thr, Phe435Leu, Lys463Asn, Phe480Leu) in Lac$^{ep69}$ have also changed.

The results of Prote-Param analysis of Lac, Lac$^{ep69}$, and D500G are shown in Table 3. First, D500G and Lac$^{ep69}$ are stable proteins. According to the stability coefficient data, Lac (40.03)>Lac$^{ep69}$ (39.57)>D500G (39.40). It is noteworthy that D500G has the highest stability. The change of the protein's primary structure will affect its spatial structure, which will affect the intermolecular forces and steric hindrance to varying degrees, thereby changing the stability of the protein. Also, the increase in stability provides theoretical support for the increase in the enzyme activity of D500G. Secondly, wild-type and mutant laccase proteins have similar molecular weights and are both hydrophilic proteins, although the hydrophilicity of Lac$^{ep69}$ and D500G is slightly decreased. Amino acid changes, before and after mutation, led to hydrophilic amino acids were replaced by hydrophobic amino acids, hydrophobic R groups to a certain extent weakened the hydrophilicity of enzyme proteins. The difference marks are highlighted in circles in Fig. 6.

SOPMA results show that Lac, Lac$^{ep69}$, and D500G have similar secondary structures, and they are mainly composed of random coils. The results are shown in Fig. 7. Compared with the wild type, the proportion of α-helix in the mutant is increased and the random coil is decreased. Lac has 61.4% random coils, 8.19% α-helix, 5.26% β-turns, and 25.15% extended chains. Lac$^{ep69}$ has 60.62% random curl, 9.36% α-helix, 4.29% β-turn, and 25.73% extended chain. D500G has 59.45% random curl, 8.58% α-helix, 5.85% β-turn, and 26.12% extended chain. Due to the change of amino acids, the proportion of α-helix increases, the random curl decreases, and other structural changes are small.

Lac, Lac$^{ep69}$, and D500G tertiary structure prediction, Swiss-Model analysis results are shown in Fig. 8. The alpha-helical of the mutant laccase Lac$^{ep69}$ is reduced by one, except

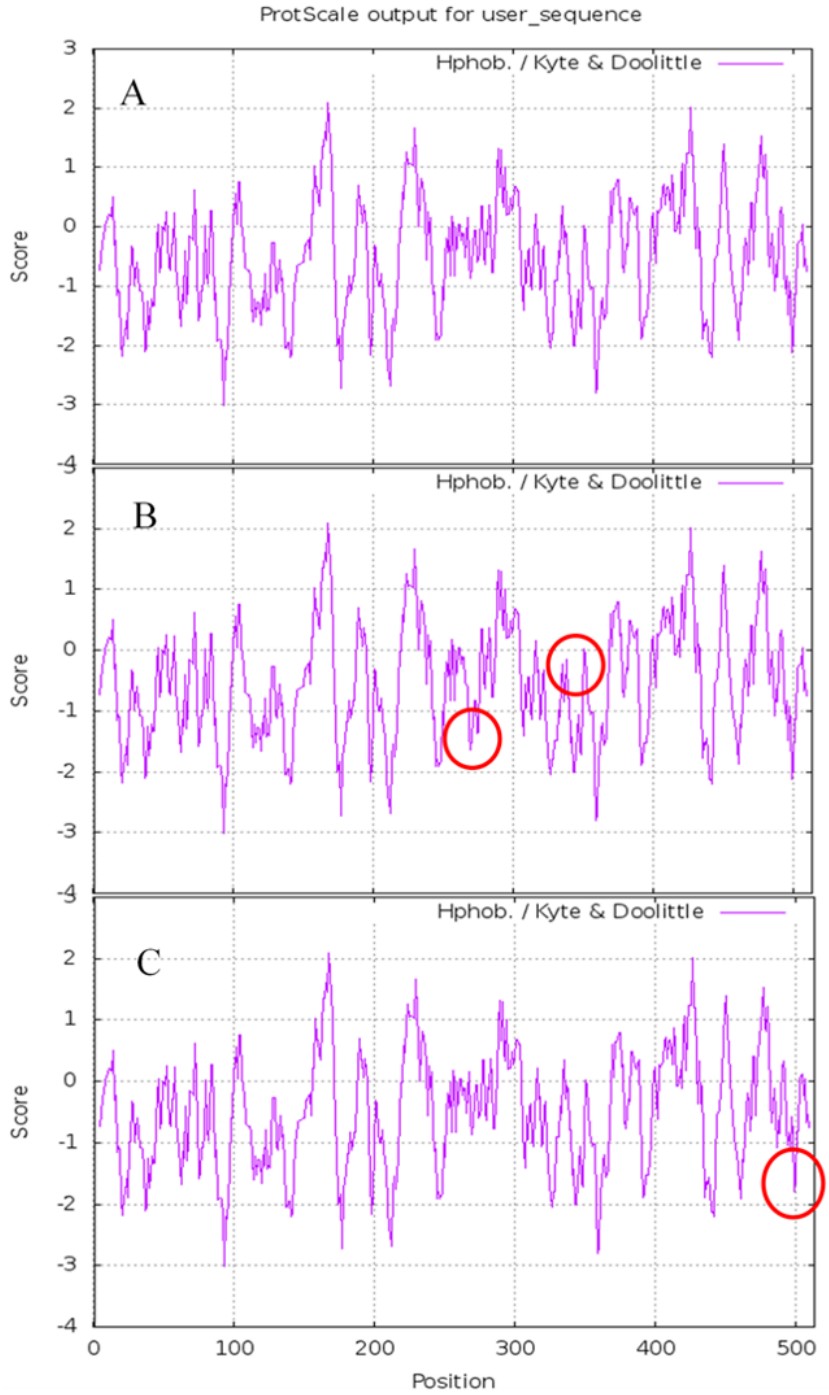

**Figure 6 Hydrophobicity of Lac, Lac$^{ep69}$, and D500G.** (A) Hydrophobicity of Lac; (B) hydrophobicity of Lac$^{ep69}$; (C) hydrophobicity of D500G.

that the amino acid at position 463 appears on the random curl, and the others are in the β-turn. The tertiary structure of Lac$^{ep69}$ is similar to Lac. However, the α-helix and β-turn angles of D500G are reduced, and the structure shows a loose transition state, in which the
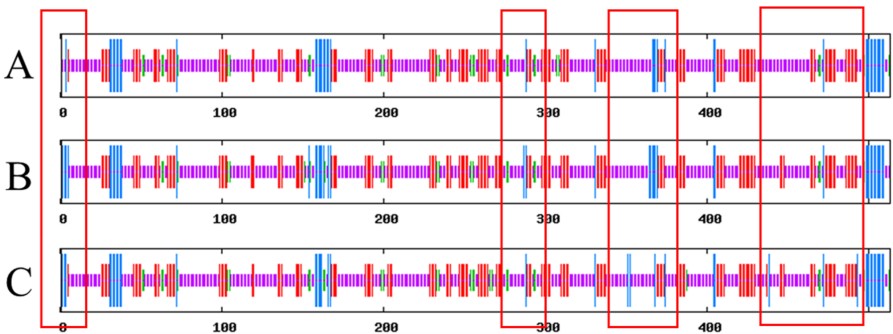

**Figure 7** **Prediction of the secondary structure of Lac, Lac[ep69], and D500G.** Blue bars show α-helix, green bars show β-turn, red bars show extended strand, purple bars show random *E. coli*. (A) Lac; (B) Lac[ep69]; (C) D500G.

amino acid at position 500 appears on the β-turn angle. The mutated base is located near the active center of the enzyme, which may increase the activity of the enzyme. Lac, Lac[ep69], and D500G mutation sites are shown in Fig. 9. Overall, the number of Lac[ep69] hydrogen bonds didn't change. Changes in amino acids affect the distribution of the atomic electron cloud around the atom. Changes in the electron cloud and hydrogen bonds may be one of the reasons affecting the enzyme activity of laccase.

## DISCUSSION

The crude laccase enzyme activity of the original *B. licheniformis* in this study was only 11.99 U/mg, which is because the endogenous expression level of most bacterial laccases is relatively low (*Chen et al., 2015*). While heterologous expression of laccase is one of the effective ways to solve this poser. The low expression level of wild-type laccase is the key to the low activity of the crude enzyme solution of *B. licheniformis*. Also, laccase belongs to intracellular localization, *E. coli* expression system is a better choice (*Wu et al., 2010*). So, we obtained a 1.37-fold increase in the expression of recombinant laccase Lac by heterologous expression in *E. coli*, but its enzyme activity was still lower. The intracellular enzyme may greatly reduce its original enzyme activity during the process of isolation and purification due to physical damage, inducer, and improper operation. Through directed evolution, two mutant laccases Lac[ep69] and D500G with improved enzyme activity were obtained. Among them, obtained by site-directed mutation has higher enzyme activity, stability, and catalytic efficiency. This result is similar to that of some *Bacillus* spp. (*Wang, Lu & Feng, 2017*; *Liu et al., 2011*).

Whether it is random mutagenesis or site-directed mutation, changes in amino acid will affect the spatial structure of the protein, which in turn affects the nature and function of the protein. Studies have shown that factors such as the proportion of certain amino acids, protein accumulation, hydrophobicity, increased helical fold content, internal hydrogen bonding and density of salt bridges, and the distribution of charged residues on the surface are important factors that affect protein thermal stability (*Kumar & Nussinov, 2001*; *Sterner & Liebl, 2001*).

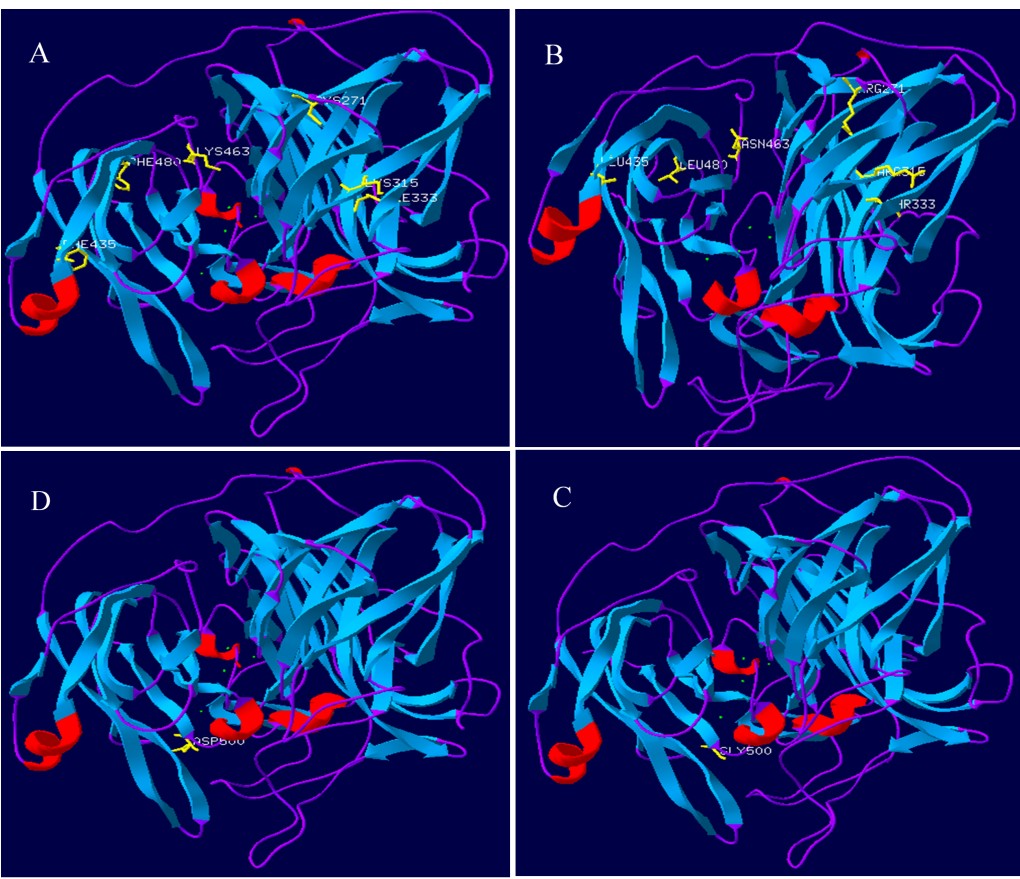

**Figure 8** **Prediction of the three-level structure prediction of Lac, Lac^ep69, and D500G.** Red represents helix; blue represents strand, purple represents coil, and yellow represents a mutant amino acid. (A, D) Lac; (B) Lac^ep69; (C) D500G.

Lac^ep69 mutated 6 amino acid positions, and cysteine at position 271 was mutated to basic arginine, which may affect the optimal pH of the enzyme. The mutation of its non-polar isoleucine at position 333 to polar threonine may affect the hydrophobicity of the protein. The change in hydrophobicity, in turn, affects the stability of the protein.

Besides, the stability of protein structure is closely related to hydrogen bonds, because 1 mol of hydrogen bonds can provide 0.6 calories of energy to maintain the stability of protein structure. The improvement of hydrogen bond introduced by the amino acid mutation at position 271 of Lac^ep69 is one of the important roles of enzyme protein stability (*Mabrouk et al., 2011*). The cysteine at position 271 and the lysine at position 315 of Lac ^ep69 are both mutated to arginine. Arginine can participate in a variety of non-covalent bond interactions, and its side chain can provide more space for charge interactions. These factors further improve the stability of the enzyme (*KnoChel et al., 1996*). D500G's acidic amino acid, aspartic acid, becomes glycine, and the change in the optimal pH may be related to the change in the polarity of the amino acid. The side chain of aspartic acid is "-$CH_2COOH$" and the side chain of glycine is hydrogen atom "-H". After the laccase

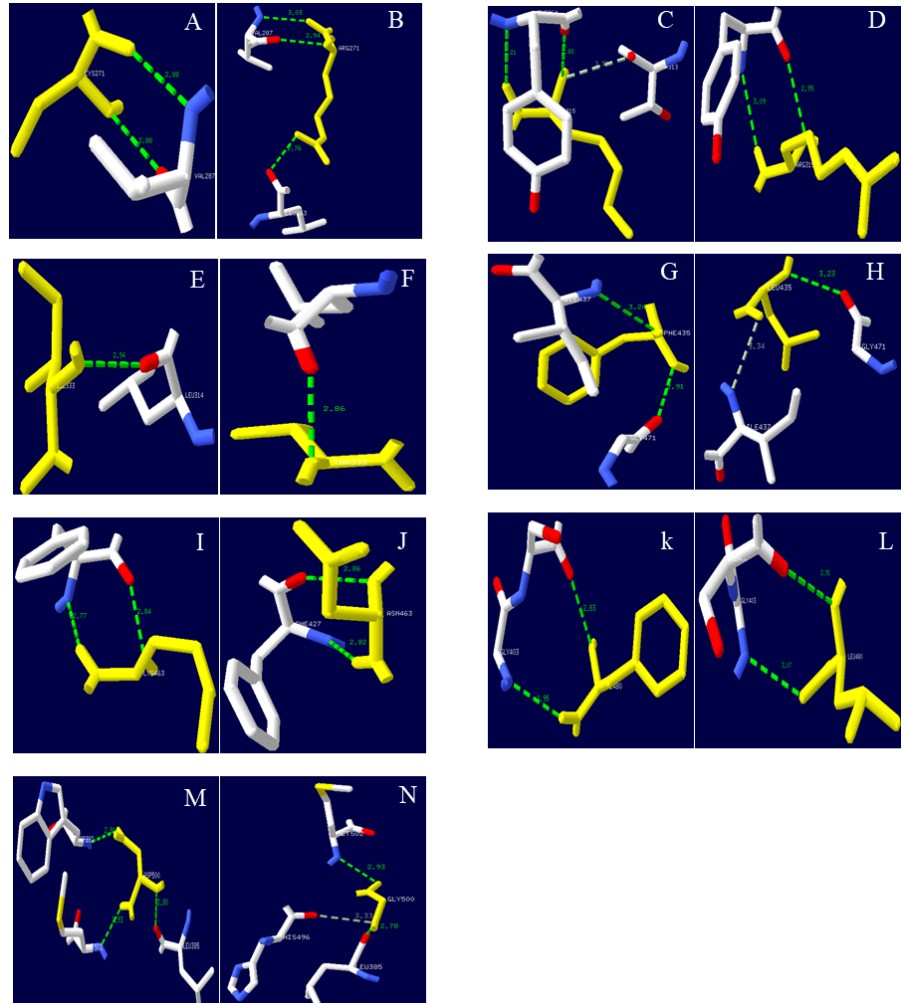

**Figure 9** **Hydrogen bond distribution of mutation sites of Lac, Lac$^{ep69}$, and D500G.** Yellow represents mutant amino acid; Green represents hydrogen bond; (A, C, E, G, I, K, M) Lac; (B, D, F, H, J, L) Lac$^{ep69}$; (N) D500G. Mutation sites are marked with red circles. Lac's Cys at position 271 and Val at position 287 form two hydrogen bonds; Lac$^{ep69}$'s 271 at Arg and Val at position 287 form two hydrogen bonds, and form a hydrogen bond with Leu at position 263. Lac's Ile at 315 and Tyr at 264 forms two hydrogen bonds, which may collide with atoms in Thr at 313; Lac$^{ep69}$'s 315 Arg and Tyr at 264 forms two hydrogen bonds. Lac's Phe at 435 forms a hydrogen bond with Ile at position 437 and Gly at position 471; Lac Lac$^{ep69}$'s Leu at position 435 forms a hydrogen bond with Ile at position 437, which may collide with an atom in Gly at position 437. Lac's aspartic acid at position 500 and leucine at position 385, tryptophan at position 392 and methionine at position 502 each form a hydrogen bond; glycine at position 500 of D500G and the methionine at position 502 each forms a hydrogen bond, and the atoms in histidine at position 496 may collide.

mutation, the flexibility of the N-C and C-C framework changes, the binding force between the enzyme and the substrate becomes stronger, and the enzyme activity improves. Changes in properties caused by the 500 amino acid change have also been found in other bacterial laccases (*Koschorreck, Schmid & Urlacher, 2009*; *Nasoohi et al., 2013*).

The secondary and tertiary structure predictions show that Lac, Lac$^{ep69}$, and D500G are similar in structure, with little difference in physical and chemical properties. In general, the mutants only form certain secondary bonds in the structure, and the distribution of the electron cloud is slightly changed, but the sites that play a key role in the function of the enzyme are not changed. It is noteworthy that D500G differs greatly from Lac in secondary structure, which may be one of the reasons for its higher thermal stability and higher enzyme activity. Because the type and number of intramolecular forces usually affect the thermal stability and catalytic activity of enzyme molecules (*Xie et al., 2014*).

After analyzing the degradation of different kinds of dyes, it is found that wild-type and mutant laccases generally have lower degradation rates for methyl orange and higher degradation rates for acid violet. From the analysis of enzyme specificity, laccase has different specificities for dye molecules of different structures, and the degree of specific binding between the enzyme and the substrate determines the degradation effect of the substrate. The anthraquinone structure of acid violet belongs to the substrate dye of bacterial laccase laccase in this study, but has a low specificity with azo methyl orange (*Yaropolov et al., 1994*; *Galai, Youssoufi & Marzouki, 2014*). From the perspective of mutants, D500G has the highest degradation rate for the three dyes among the three enzymes. The reasons for this result come from three aspects. First, from the molecular level, the mutation site of D500G structurally enhances its stability, which in turn affects its enzyme activity; second, the structural change affects the affinity between the enzyme and the substrate, and $K$m value is powerful evidence. According to the enzyme kinetic theory, the $K$m value is negatively correlated with the substrate affinity, and among the three, the $K$m value of D500G is the smallest, which is $10.50 \pm 0.32$ mM.

From the temperature analysis, the optimal temperature of the mutant is significantly improved, and high-temperature tolerance is also significantly increased. D500G still preserves more than 85% of the enzyme activity after being stored at a high temperature of $50-80\,°C$ for one hour, which provides strong evidence for the improvement of the stability of the mutant. From the perspective of pH analysis, the optimum pH of D500G becomes larger, and at the same time, it still retains more than 80% of the enzyme activity after being stored at 4.0–5.5 for one hour. The improvement of temperature and pH tolerance increased the tolerance of D500G in dyes, and at the same time reducing the impact of the environment on enzyme activity, thereby increasing the mutant's degradation rate of dyes. The results of Figs. 5A–5C also proved that the mutant has a higher degradation rate in the first hour of the dye degradation process, and tends to end after 1 h. It is speculated that the enzyme activity of the laccase in 1 h is due to various reasons. The reduction ultimately leads to a decrease in its degradation rate.

## CONCLUSION

In this study, we successfully screened a mutant laccase D500G with significantly improved decolorization of dyes by site-directed mutagenesis. Compared with wild-type laccase, D500G has significantly improved temperature and pH stability, which further enhances the tolerance of laccase in the dye wastewater environment and is more suitable for practical industrial production.

### Funding
The authors received no funding for this work.

### Competing Interests
The authors declare there are no competing interests.

### Author Contributions
- Tongliang Bu conceived and designed the experiments, analyzed the data, prepared figures and/or tables, authored or reviewed drafts of the paper, and approved the final draft.
- Rui Yang conceived and designed the experiments, performed the experiments, prepared figures and/or tables, and approved the final draft.
- YanJun Zhang and Yuntao Cai performed the experiments, prepared figures and/or tables, and approved the final draft.
- Zizhong Tang, Chenglei Li, Qi Wu and Hui Chen analyzed the data, authored or reviewed drafts of the paper, and approved the final draft.

### Data Availability
The raw measurements are available in the Supplemental Files.

### Supplemental Information
Supplemental information for this article can be found online at http://dx.doi.org/10.7717/peerj.10267#supplemental-information.

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
