# Peer review of "Improving decolorization of dyes by laccase from Bacillus licheniformis by random and site-directed mutagenesis"

_PeerJ, doi:10.7717/peerj.10267_

## Round 0.1 · original submission · Major Revisions

Please pay special attention to the points a-e raised by the Reviewer 2

Reviewer 1 ·

Basic reporting

In the abstract section, I suggest giving information about the origin of the enzyme, at least mentioning the organism.

Line 27. It is not necessary to write that it has been repeatedly verified.


In the introduction section.
I suggest a bit more information on the origin of the enzyme, there is very little mention of the organism from which the enzyme comes from and although it is not the objective of the investigation, I consider it important to add information so that the audience knows the organism that produces said enzyme.

Line 66. The organism has not been mentioned before, write the full name.

I consider that line 10 of the abstract can be developed a little more in the introduction section; This will give the study more impact.

Experimental design

Line 77: acid, acid??

Line 79: Eco RI

Line 93-98: Method reference

Line 112-113: Method reference or if it is from this study, I suggest describing it a bit more, for reproducibility.

Line 127. You can save the full name of the IPTG (it was already described above in the text)

Validity of the findings

Figure 1, I suggest placing it in the supplementary material.

Figs 12 and 13. Mention in the legend of the figure, the development method (coomassie or silver).

Lines 231 and 232: Here are the repeated data, I suggest removing them, as they are the same from the previous lines.


Fig 6.
The legend in figure 6 is confusing. I suggest Reorder the indications to observe and analyze the figure.

Line: 254. There are no results in this table, instead are the primers for the PCR. Please, check this part.


In Figure 8. Specifically D500G, you know the location of the amino acids, so it would be helpful if in this figure you could point them out, at least with a box and observe that possible changes; showed in fig 9.


Line: 285. Bacillus licheniformis  B. licheniformis.



In the supplementary material (specifically: optimum copper ion concentration) you should switch to the English language.


In the conclusion section. I suggest not to expand too much and not to put data that was previously shown and discussed in the text.

Additional comments

Only if you have images of the dyes with the respective enzymes, wild type and mutant; They would be of great help in observing the improvement in discoloration and the laccase activities. It is just a suggestion and if you have the pictures you could include them in the supplementary material.

Reviewer 2 ·

Basic reporting

No comment.

Experimental design

No comment.

Validity of the findings

No comment.

Additional comments

This is an interesting study in the field of environmental biotechnology. The paper is generally well structured and underlines the usefulness of random and site-directed mutagenesis in improving enzyme activity, stability, catalytic efficiency and dye decolorization ability of bacterial laccases. However, in my opinion this paper has some shortcomings in regards to some data analyses. Below I have provided numerous remarks.
Key critical points are : a) a lack of a discussion of magnetic and spectral properties of these true or laccase-like multi-copper oxidases compared to other bacterial laccases, b) a lack of zymograpgy data revealing enzyme activity and dye decolorization potential, c) a lack of statistical optimization approach using experimental designs for dye decolourization process, d) the authors should give proper and more justification for their claim that the mutant enzyme has a high degradation capacity for the dyestuffs. For this, in-silico, in-vitro or in-vivo toxicity tests of dye degradation products should be given. e) the activity of mutant enzyme at acidic pHs could limit its application in dye wastewater treatment particularly in textile industry, and f) finally, many typos and grammer check up is also needed (lines 52, 293, 306, ..).
Given these shortcomings the manuscript requires major revisions.

---

## Round 0.2 · accepted · Accept

The manuscript is accepted for publication in PeerJ.

Reviewer 1 ·

Basic reporting

Thank you very much for attending to the proposed suggestions.

Experimental design

Thank you very much for attending to the proposed suggestions.

Validity of the findings

Thank you very much for attending to the proposed suggestions.

Additional comments

Thank you very much for attending to the proposed suggestions. I have no other observation.